**Data Availability Statement:** All relevant data are within the manuscript and its Supporting Information files. The gene datasets applied in this study are also publicly available from the Gene

# Identification of driver genes and key pathways of non-functional pituitary adenomas predicts the therapeutic effect of STO-609

**Bo Wu**[1,2ᵒ], **Shanshan Jiang**[3ᵒ], **Xinhui Wang**[1,4], **Sheng Zhong**[5], **Yiming Bi**[6], **Dazhuang Yi**[6], **Ge Liu**[7], **Fangfei Hu**[7], **Gaojing Dou**[1,8], **Yong Chen**[6], **Yi Wu**[9‡]*, **Jiajun Dong**[9‡]*

1 Clinical College, Jilin University, Changchun, China, 2 Department of Orthopedics, Jilin University First Hospital, Changchun, China, 3 Institute of Zoology, China Academy of Science, Beijing, China, 4 Department of Oncology, Jilin University First Hospital, Changchun, China, 5 Department of Neurosurgery, Cancer Hospital of Sun Yat sen University, Guangzhou, China, 6 Department of Neurosurgery, The First Bethune Hospital of Jilin University, Changchun, China, 7 College of Pharmacy, Jilin University, Changchun, China, 8 Department of Breast Surgery, Jilin University First Hospital, Changchun, China, 9 Department of Neurosurgy, Jiangmen Central Hospital, Jiangmen, China

ᵒ These authors contributed equally to this work.
‡ YW and JD also contributed equally to this work.
* wuyi_jiangmen@163.com (YW); dongjj_jiangmen@163.com (JD)

## Abstract

### Objective

Our study is to identify DEGs (Differentially Expressed Genes), comprehensively investigate hub genes, annotate enrichment functions and key pathways of Non-functional pituitary adenomas (NFPAs), and also to verify STO-609 therapeutic effect.

### Methods

The gene expression level of NFPA and normal tissues were compared to identify the DEGs (Differential expressed genes) based on gene expression profiles (GSE2175, GSE26966 and GSE51618). Enrichment functions, pathways and key genes were identified by carrying out GO (Gene Ontology), KEGG (Kyoto Encyclopedia of Genes and Genomes) analysis and PPI (Protein-Protein Interation) network analysis. Moreover, experiments in vitro were conducted to verify the anti-NFPAs effects of STO-609.

### Results

169 over-expression genes and 182 low expression genes were identified among 3 datasets. Dopaminergic synapse and vibrio cholerae infection pathways have distinctly changed in NFPA tissues. The Ca$^{2+}$/CaM pathway played important roles in NFPA. Four hub proteins encoded by genes *CALM1*, *PRDM10*, *RIPK4* and *MAD2L1* were recognized as hub proteins. In vitro, assays showed that STO-609 induced apoptosis of NFPA cells to inhibit the hypophysoma cellular viability, diffusion and migration.

Expression Omnibus repository (http://www.ncbi.nlm.nih.gov/geo).

**Funding:** This study was supported by the Science and Technology Planning Project of Jiangmen, China (20186301001100119805).

**Competing interests:** The authors have declared that no competing interests exist.

**Abbreviations:** BP, Biological processes; CALM1, Calmodulin 1; CaM-KK, Calmodulin-dependent protein kinase kinase; CC, Cell component; CCK-8, Cell Counting Kit-8; DAVID, Database for Annotation, Visualization and Intergrated Discovery; DEG, Differential expressed genes; DMEM, Dulbecco's modified Eagle's medium; FBS, Fetal bovine serum; GEO, Gene Expression Omnibus; GO, Gene ontology; GSEA, Gene Set Enrichment Analysis; KEGG, Kyoto encyclopedia of Genes and Genomes; MAD2L1, Mitotic arrest deficient 2 like 1; MF, Molecular function; NFPAs, Non-functional pituitary adenomas; NS, Normal saline; PA, Pituitary adenomas; PCA, Principal component analysis; PD, Parkinson's disease; PRDM10, PR/SET domain 10; PRL, Prolactin; PPI, Protein-protein interation; RIPK4, Receptor interacting serine/threonine kinase 4; STRING, Search Tool for Retrieval of Interacting Genes; WHO, World Health Organization.

## Conclusion

Four hub proteins, encoded by gene *CALM1*, *PRDM10*, *RIPK4* and *MAD2L1*, played important roles in NFPA development. The Ca$^{2+}$/CaM signaling pathway had significant alternations during NFPA forming process, the STO-609, a selective CaM-KK inhibitor, inhibited NFPA cellular viability, proliferation and migration. Meanwhile, NFPA was closely related to parkinson's disease (PD) in many aspects.

## Introduction

Human pituitary adenomas, accounting for 10% of intracranial tumors, are common intracranial primary neoplasms [1]. Pituitary adenomas (PA) encompass two types, one is functional endocrine, symptomatic or functional active type, and the other one is non-functioning, null cell or functionally inactive type [2]. Non-functional Pituitary Adenomas (NFPAs) comprises approximately 30% of PA [3]. The levels of hormones in the blood do not alter significantly in NFPA patients, accordingly the patients don't have any clinical symptoms caused by hormone hypersecretion. Although NFPAs are histologically benign tumors [4], which is defined as level I according to WHO (2016) pathologic grading criteria [5], it is difficult to diagnose them in the primary stage due to the lack of clinical manifestations of hormone hypersecretion and the absence of specific serological markers. Along with the condition development, tumors suppress adjacent tissue and then clinical symptoms appear, which accounts for patients compression symptoms [6]. These complications will seriously affect the nervous system function and reduce the quality of life. In addition, the overall risk of malignant tumors in patients with NFPA is higher [7].

The diagnostic approaches regarding to NFPAs include perimetry, the evaluation of all anterior pituitary hormone systems, and the sellar region of tumor in MRI [8]. Except for a few functional PA, which can be controlled by drugs, it hasn't been any effective drug treatment for NFPAs so far. Treatments for NFPAs are surgical treatment and radiotherapy. The standard treatment for NFPA is surgical resection, endoscopic trans-sphenoidal approach is the most accepted approach to reduce the tumor size and to improve clinical symptoms [9, 10]. NFPAs are mostly benign, however, the tumor sometimes suppresses and invades the surrounding tissue. Thus tumors are difficult to be resected completely and tend to relapse after initial surgery [11]. The recurrence rate after operation is up to 30%. Radiation therapy is usually used in the treatment of persistent or recurrent adenomas, however, the treatment effect is still unsatisfactory [8]. So novel therapeutic methods and medicines are needed urgently.

After decades studies, the molecular pathogenesis of NFPAs is still turbid. Our knowledge regarding to NFPAs is still superficial and provincial [12], there is a lack of definite index to guide the clinical practice. In addition, trans-sphenoidal surgery, the most accepted treatment approach for NFPAs, still contains many surgical complications, such as postoperative endocrine deficits, postoperative infection, cerebrospinal fluid leak etc. However, there is still no potent drug for NFPAs. The genes, proteins interaction and small molecules drugs research regarding to NFPAs have been rarely conducted in the past decades. So conducting a study which comprehensively depicts molecular pathogenesis of NFPAs as well as identifies novel drugs for NFPAs is necessary and crucial.

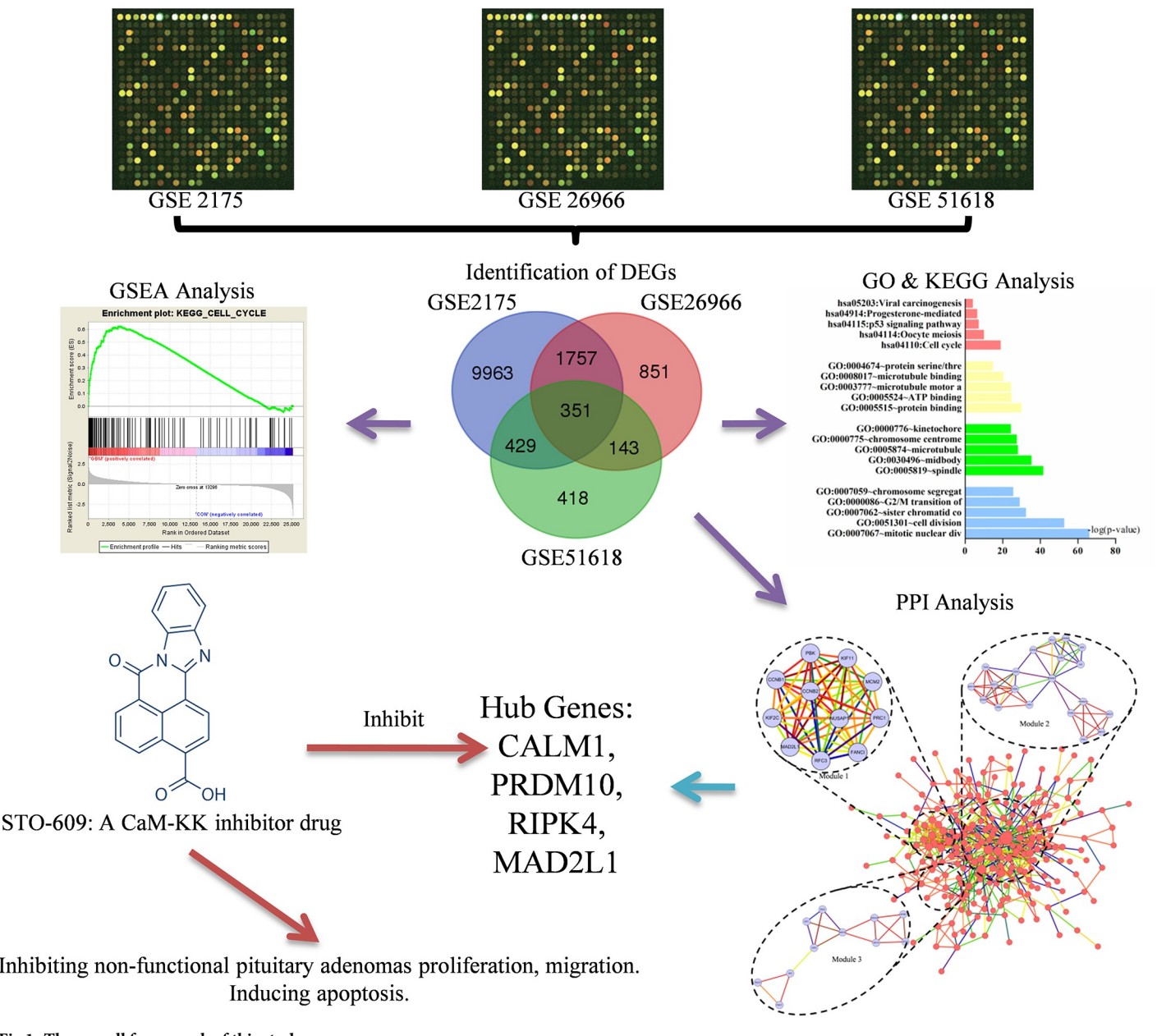

**Fig 1. The overall framework of this study.**

Our study used bioinformatics methods, especially combining with microarray technology, GO, KEGG analysis and PPI network analysis, employing 3 datasets (GSE2175, GSE26966 and GSE51618) to screen out the DEGs. Then, hub genes and key pathways were screened, which could be exploited to select novel molecular targets for NFPA diagnosis and treatment. Meanwhile, a series of assays in vitro were conducted to verify the anti-NFPAs effect of STO-609, which is a selective and cell-permeable inhibitor of the Ca2+/calmodulin-dependent protein kinase kinase (CaM-KK) [13]. STO-609 operates downstream of CALMA, and inhibits the Ca$^{2+}$/CaM signaling pathway on account of the ATP-competitive inhibition to CaM-KK [14], as shown in Fig 1.

## Materials and methods

### Microarray data

Gene Expression Omnibus (GEO, http://www.ncbi.nlm.nih.gov/geo) database is a public database, which stores a great quantity of studies related to high throughput functional genomic researches [15]. Profiles GSE2175, GSE26966 and GSE51618 were obtained from this database. The GSE2175 dataset contains 5 samples, from which we choose a normal one and a pathological one [16]. The GSE26966 dataset contains 23 samples, from which we choose 13 effective ones (9 normal and 4 pathological) [17]. And the GSE51618 dataset contains 3 normal samples and 7 pathological samples.

### Identification of DEGs

Genespring software (version 11.5, Agilent, USA) was used for data analysis. We compared expression level of genes of NFPAs and normal tissue for DEGs. In GeneSpring, hierarchical clustering analysis and principal component analysis (PCA) were applied to ensure probe quality control. Probes that intensity values below 20th percentile were eliminated using the "filter probesets by expression" option. Classical t test was performed with a change > 2 fold and $P < 0.05$ was defined to be statistically significant. For three datasets, we did the same operation three times. Then, website tool was used to draw Venn diagrams of three groups of DEGs. (http://bioinformatics.psb.ugent.be/webtools/Venn/).

### Gene ontology and pathway enrichment analysis

The Database for Annotation, Visualization and Integrated Discovery (DAVID, http://david.abcc.ncifcrf.gov/) provides tools to help users learn the biological meaning behind DEGs. Biological processes (BP), molecular functions (MF) and cellular components (CC) of genes were analyzed with GO analysis, and enrichment pathways of those genes were identified with KEGG analysis.

### Integration of PPI network construction and modules selection

STRING (Search Tool for the Retrieval of Interacting Genes, http://string.embl.de/) database provides an available option for users to evaluate the proteins interaction information on line. Then, the information was analyzed in Cytoscape software. Main modules were screened with scores > 3.6 and number of nodes>11 by Molecular Complex Detection (MCODE). Genes belong to modules were also performed with GO and KEGG analysis.

### Cell lines and reagents

HP75, a non-functional pituitary adenoma cell line, was cultured in DMEM (BioWhittaker, Cambrex Corp., Nottingham, UK) containing 15% horse serum (TCS Cellworks, Buckingham, UK), and 2.5% fetal calf serum (Life Technologies, Paisley, UK) at 37˚C in an easeful air atmosphere containing 5% carbon dioxide. AtT-20 (mouse normal pituitary cells) and GT1-1 (mouse pituitary adenoma cells) were cultured in DMEM, containing 10% FCS, at 37˚C in the same air environment. STO-609, a selective inhibitor of CaM-KK, which inhibited the $Ca^{2+}$/CaM signaling pathway, was purchased from Apexbio Inc. (Apexbio, Houston, USA). STO-609 was dissolved in DMSO to obtain the stock solution, then appropriate culture medium was respectively added to the stock solution to configure cell culture medium with different STO-609 concentration.

## CCK-8 assay

The cells, HP75 and GT1-1, viability were assessed by Cell Counting Kit-8 (CCK-8) (Dojindo Laboratories, Kumamoto, Japan). We seeded the cells into 96-well plates for overnight, and the density is $1.0 \times 10^5$ cells/well. After washing culture medium, different doses of STO-609 were used to the cells and cultured for 24h, control group and solvent control group were classed by using normal saline (NS) and DMSO respectively, and 6 wells were prepared for doses of STO-609 (concentration gradients were 0.4μmol/L, 0.8μmol/L, 1.6μmol/L, 3.2μmol/L, 6.4μmol/L, 12.8μmol/L, 25.6μmol/L and 51.2μmol/L). Cells were cultured for 1h after added CCK-8 into wells with a quantity of 10μl/well. The wave length of 450 nm was applied to measure OD value of each well on the microplate reader (Multiskan, Thermo, USA).

## Colony-forming assay

HP75 and GT1-1 were inoculated in a six-well cell culture plate with density of 50 cells per square centimetre, and the surface area of each well of the culture plate was 9.6 cm$^2$. After 24h in culture, we configured cell culture medium with STO-609 concentration of 0.25μmol/L and 1μmol/L. The concentration of DMSO was less than 0.1%. In this concentration, the influence of DMSO on cells was negligible. After 10 days, we counted and described colonies refer to Franken et al. [18]. Moreover, colonies were dyed by 5% crystal violet for half an hour after fixed in 4% paraformaldehyde.

## In vitro scratch assay

GT1-1 cells were seeded into PermanoxTM plates (24-well) and cultured. The cell-free area was made with a 1ml pipette. After 24h in culture, different doses of STO-609 were used to treat the cells at 0, 12, 24 hours, we captured the scraped area images with phase contrast microscopy, and measured the wounds and scratch width.

## Apoptosis assays

The HP75 cells in the log growth phase were inoculated into 6-well plates, and the density was $2 \times 10^5$ cells/well. Different doses of STO-609 were used to treat cells. After 24h in culture, cells were harvested and Annexin-V-FITC/PI labeling was conducted. Then, the stained cells were analyzed flow cytometer and calculated with FACSDiva version 6.2.

## Statistical analysis

SPSS18.0(SPSS Inc., Chicago, Illinois, USA) was applied for all statistics data. The independent-samples t test method was used for quantitative data of bioinformatical analysis, while Analysis of Variance (ANOVA) was conducted to analyse multiple comparison data of cell assays. Dunnett-t test was performed as post hoc test after ANOVA. Significance level was marked with P values $< 0.05$.

# Results

## Identification of DEGs

There are 12500, 3102 and 1341 DEGs were identified from GSE2175, GSE26966, and GSE51618 profiles. Then Venn plot showed that 351 DEGs in NFPA tissues in all three datasets, among which were 169 up-regulated and 182 down-regulated. The Venn plot of DEGs was exhibited on Fig 2A, and the details were placed on S1 Table. The hub genes expression heat maps were also exposed on Fig 2C and 2D.

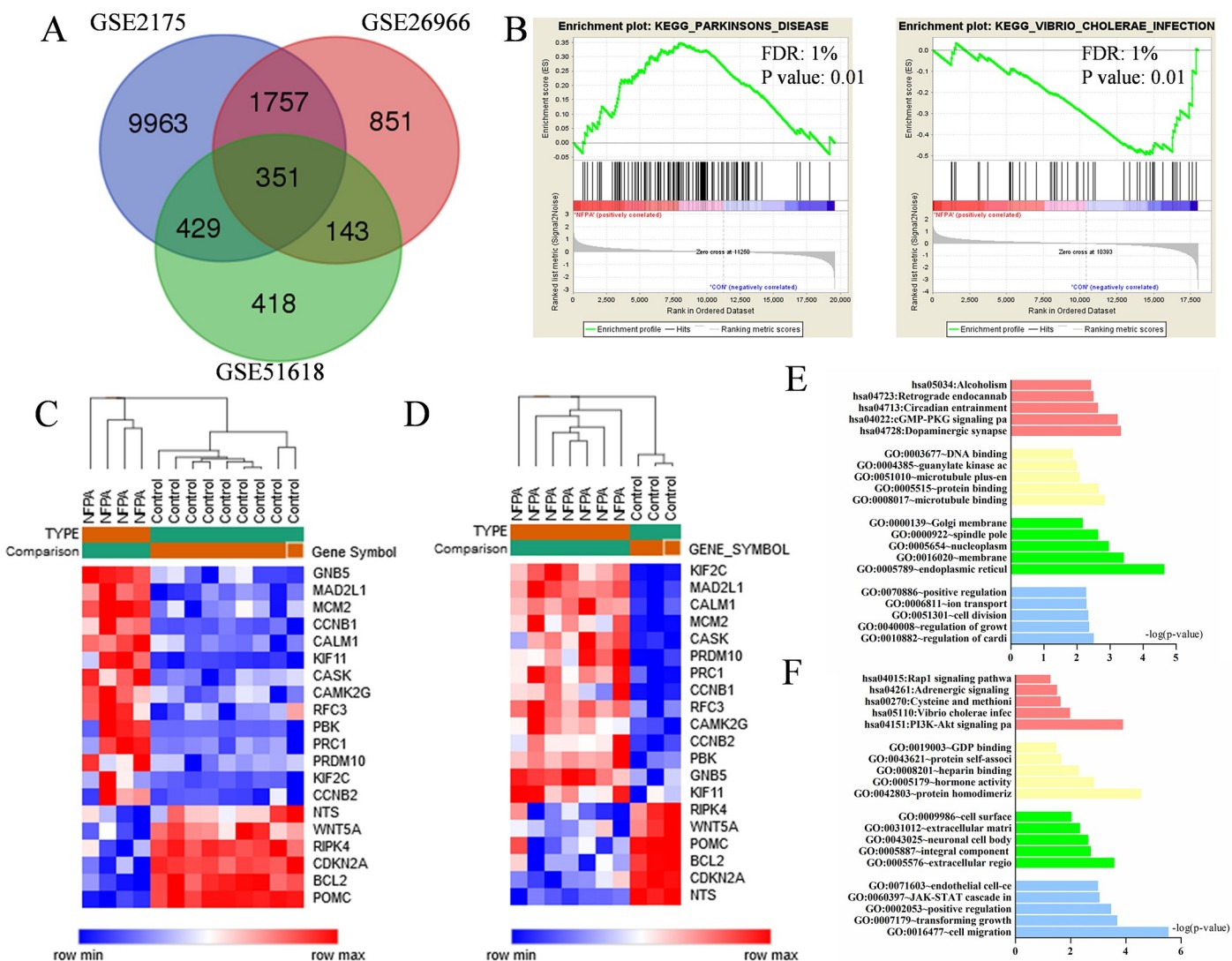

**Fig 2. (A)** The Venn plot of DEGs among three datasets **(B)** GSEA analysis results **(C)** Hub gene expression heat map of GSE26966 **(D)** Hub gene expression heat map of GSE51618 **(E)** Functional and pathway enrichment analysis of up-regulated genes among three datasets **(F)** Functional and pathway enrichment analysis of down-regulated genes among three datasets.

## Gene ontology and pathway enrichment analysis of DEGs

In order to obtain further insight of DEGs, we respectively put the up-regulated and down-regulated DEGs into DAVID, and the details of results were showed in Fig 2B, 2E and 2F and Table 1. GO analysis results demonstrated that up-regulated DEGs were enriched in regulation of myocardial contraction by calcium signal and regulation of growth in BP analysis, while CC analysis displayed the enrichment in endoplasmic reticulum membrane, membrane and nucleoplasm. As for MF analysis, microtubule binding and protein binding are enriched with up-regulated DEGS. Meanwhile, the down-regulated DEGs in BP analysis were enriched in cell migration and transforming growth factor β receptor signaling pathway. Besides, the CC analysis showed the enrichment of these down-regulated DEGs in extracellular region, integral component of plasma membrane and neuronal cell body. For MF analysis, the enrichment

**Table 1. Functional and pathway enrichment analysis of up-regulated and down-regulated genes in NFPAs.**

| Expression | Category | Term | Count | % | PValue |
|---|---|---|---|---|---|
| Up-regulated | GOTERM_BP_DIRECT | GO:0010882~regulation of cardiac muscle contraction by calcium ion signaling | 3 | 1.29 | 3.11E-03 |
| | GOTERM_BP_DIRECT | GO:0040008~regulation of growth | 5 | 2.15 | 4.24E-03 |
| | GOTERM_BP_DIRECT | GO:0051301~cell division | 12 | 5.15 | 4.52E-03 |
| | GOTERM_BP_DIRECT | GO:0006811~ion transport | 7 | 3.00 | 5.14E-03 |
| | GOTERM_BP_DIRECT | GO:0070886~positive regulation of calcineurin-NFAT signaling cascade | 3 | 1.29 | 5.24E-03 |
| | GOTERM_CC_DIRECT | GO:0005789~endoplasmic reticulum membrane | 27 | 11.59 | 2.26E-05 |
| | GOTERM_CC_DIRECT | GO:0016020~membrane | 46 | 19.74 | 3.83E-04 |
| | GOTERM_CC_DIRECT | GO:0005654~nucleoplasm | 53 | 22.75 | 1.08E-03 |
| | GOTERM_CC_DIRECT | GO:0000922~spindle pole | 7 | 3.00 | 2.28E-03 |
| | GOTERM_CC_DIRECT | GO:0000139~Golgi membrane | 16 | 6.87 | 6.66E-03 |
| | GOTERM_MF_DIRECT | GO:0008017~microtubule binding | 10 | 4.29 | 1.42E-03 |
| | GOTERM_MF_DIRECT | GO:0005515~protein binding | 134 | 57.51 | 2.21E-03 |
| | GOTERM_MF_DIRECT | GO:0051010~microtubule plus-end binding | 3 | 1.29 | 8.23E-03 |
| | GOTERM_MF_DIRECT | GO:0004385~guanylate kinase activity | 3 | 1.29 | 9.80E-03 |
| | GOTERM_MF_DIRECT | GO:0003677~DNA binding | 33 | 14.16 | 1.29E-02 |
| | KEGG_PATHWAY | hsa04728:Dopaminergic synapse | 9 | 3.86 | 4.65E-04 |
| | KEGG_PATHWAY | hsa04022:cGMP-PKG signaling pathway | 10 | 4.29 | 5.83E-04 |
| | KEGG_PATHWAY | hsa04713:Circadian entrainment | 7 | 3.00 | 2.31E-03 |
| | KEGG_PATHWAY | hsa04723:Retrograde endocannabinoid signaling | 7 | 3.00 | 3.15E-03 |
| | KEGG_PATHWAY | hsa05034:Alcoholism | 9 | 3.86 | 3.73E-03 |
| Down-regulated | GOTERM_BP_DIRECT | GO:0016477~cell migration | 13 | 5.33 | 2.85E-06 |
| | GOTERM_BP_DIRECT | GO:0007179~transforming growth factor beta receptor signaling pathway | 8 | 3.28 | 2.05E-04 |
| | GOTERM_BP_DIRECT | GO:0002053~positive regulation of mesenchymal cell proliferation | 5 | 2.05 | 3.42E-04 |
| | GOTERM_BP_DIRECT | GO:0060397~JAK-STAT cascade involved in growth hormone signaling pathway | 4 | 1.64 | 8.98E-04 |
| | GOTERM_BP_DIRECT | GO:0071603~endothelial cell-cell adhesion | 3 | 1.23 | 1.01E-03 |
| | GOTERM_CC_DIRECT | GO:0005576~extracellular region | 38 | 15.57 | 2.59E-04 |
| | GOTERM_CC_DIRECT | GO:0005887~integral component of plasma membrane | 32 | 13.11 | 1.84E-03 |
| | GOTERM_CC_DIRECT | GO:0043025~neuronal cell body | 12 | 4.92 | 2.28E-03 |
| | GOTERM_CC_DIRECT | GO:0031012~extracellular matrix | 11 | 4.51 | 4.50E-03 |
| | GOTERM_CC_DIRECT | GO:0009986~cell surface | 15 | 6.15 | 9.15E-03 |
| | GOTERM_MF_DIRECT | GO:0042803~protein homodimerization activity | 25 | 10.25 | 2.80E-05 |
| | GOTERM_MF_DIRECT | GO:0005179~hormone activity | 7 | 2.87 | 1.40E-03 |
| | GOTERM_MF_DIRECT | GO:0008201~heparin binding | 8 | 3.28 | 4.92E-03 |
| | GOTERM_MF_DIRECT | GO:0043621~protein self-association | 4 | 1.64 | 2.17E-02 |
| | GOTERM_MF_DIRECT | GO:0019003~GDP binding | 4 | 1.64 | 3.29E-02 |
| | KEGG_PATHWAY | hsa04151:PI3K-Akt signaling pathway | 17 | 6.97 | 1.25E-04 |
| | KEGG_PATHWAY | hsa05110:Vibrio cholerae infection | 5 | 2.05 | 1.04E-02 |
| | KEGG_PATHWAY | hsa00270:Cysteine and methionine metabolism | 4 | 1.64 | 2.32E-02 |
| | KEGG_PATHWAY | hsa04261:Adrenergic signaling in cardiomyocytes | 7 | 2.87 | 3.09E-02 |
| | KEGG_PATHWAY | hsa04015:Rap1 signaling pathway | 8 | 3.28 | 5.38E-02 |

was in protein homodimerization activity. In addition, the KEGG analysis results respectively demonstrated the enriched pathways of up- and down-regulated DEGs: dopaminergic synapse was where up-regulated DEGs mainly enriched, while vibrio cholerae infection was the point of down-regulated DEGs. The GSEA analysis results revealed that the pathway of parkinson's disease (PD) and the pathway of vibrio choleras infection were significantly altered.

**Table 2. Top 20 hub genes which were screened with degrees more than 12.**

| Gene symbol | Degree | Betweenness Centrality | Gene symbol | Degree | Betweenness Centrality |
|---|---|---|---|---|---|
| PRDM10 | 57 | 0.42790519 | KIF11 | 16 | 0.01211223 |
| RIPK4 | 33 | 0.19695202 | PBK | 16 | 0.03231519 |
| CALM1 | 28 | 0.10316316 | KIF2C | 15 | 0.01750419 |
| BCL2 | 28 | 0.09331486 | PRC1 | 15 | 0.00761907 |
| CCNB1 | 22 | 0.07022771 | CAMK2G | 14 | 0.02860711 |
| CDKN2A | 19 | 0.03962583 | RFC3 | 14 | 0.02189282 |
| POMC | 18 | 0.0389501 | WNT5A | 13 | 0.05815162 |
| MAD2L1 | 17 | 0.00857124 | GNB5 | 13 | 0.04135738 |
| MCM2 | 17 | 0.02005968 | NTS | 13 | 0.06497129 |
| CCNB2 | 16 | 0.01160158 | CASK | 13 | 0.03506031 |

## Integration of PPI network construction and modules selection

The hub nodes with degrees more than 12 were defined as hub proteins, which played important roles in NFPA listing in Table 2. Four hub proteins encoded by genes *CALM1* (calmodulin 1), *PRDM10* (PR/SET domain 10), *RIPK4* (receptor interacting serine/threo-nine kinase 4) and *MAD2L1* (mitotic arrest deficient 2 like 1), played significant roles in NFPA. Among these proteins with high node degree in NFPAs, PRDM10 got the highest degree of 57. Based on the PPI network of DEGs, a notable module was screened out through MCODE, including 349 nodes and 594 edges with a 3.4 average node degree, which was shown in Fig 3, and the functional annotation and enrichment of modules genes was listed respectively in Table 3. Functions of genes in module 1 were mainly enriched in spindle pole, cell division and microtubule cytoskeleton; in module 2, the enrichment showed primarily in cellular response to hypoxia, response to drug and pathways in cancer; and in module 3, genes were mainly correlated to Wnt signaling pathway and beta-catenin destruction complex disassembly.

## STO-609 reduced proliferation of NFPA cells

CAM pathway activated PI3K which stimulated important signaling pathways of NFPAs as reported. Therefore, CALM1 was regarded as the vital therapeutic target for NFPAs. To estimate effects of STO-609 in NFPA cells, CCK-8 assay and colony-forming assay were performed. As Fig 4A showed, the cellular viability of AtT-20 cells was slowly declined following the augment of drug concentrations, and HP75 and GT1-1 groups decreased more significantly compared to AtT-20 group. The results of colony-forming assay pointed that compared with the control group, the clonogenicities in petri dish treated by STO-609 were less and smaller (Fig 4B). The numbers of clone formation in drug groups was lower than in control group significantly (Fig 4D and 4E).

## STO-609 inhibits migration of NFPA cells

To evaluate migration of GT1-1 cell lines, the widths of cell-free area were measured. After 24h, the results exposed that the widths of scratch in control group were thinner than that in STO-609 group ($P < 0.05$). Meanwhile, the wounds in the drug group were quite wider than in control group after 48h ($P < 0.05$) (Fig 4C).

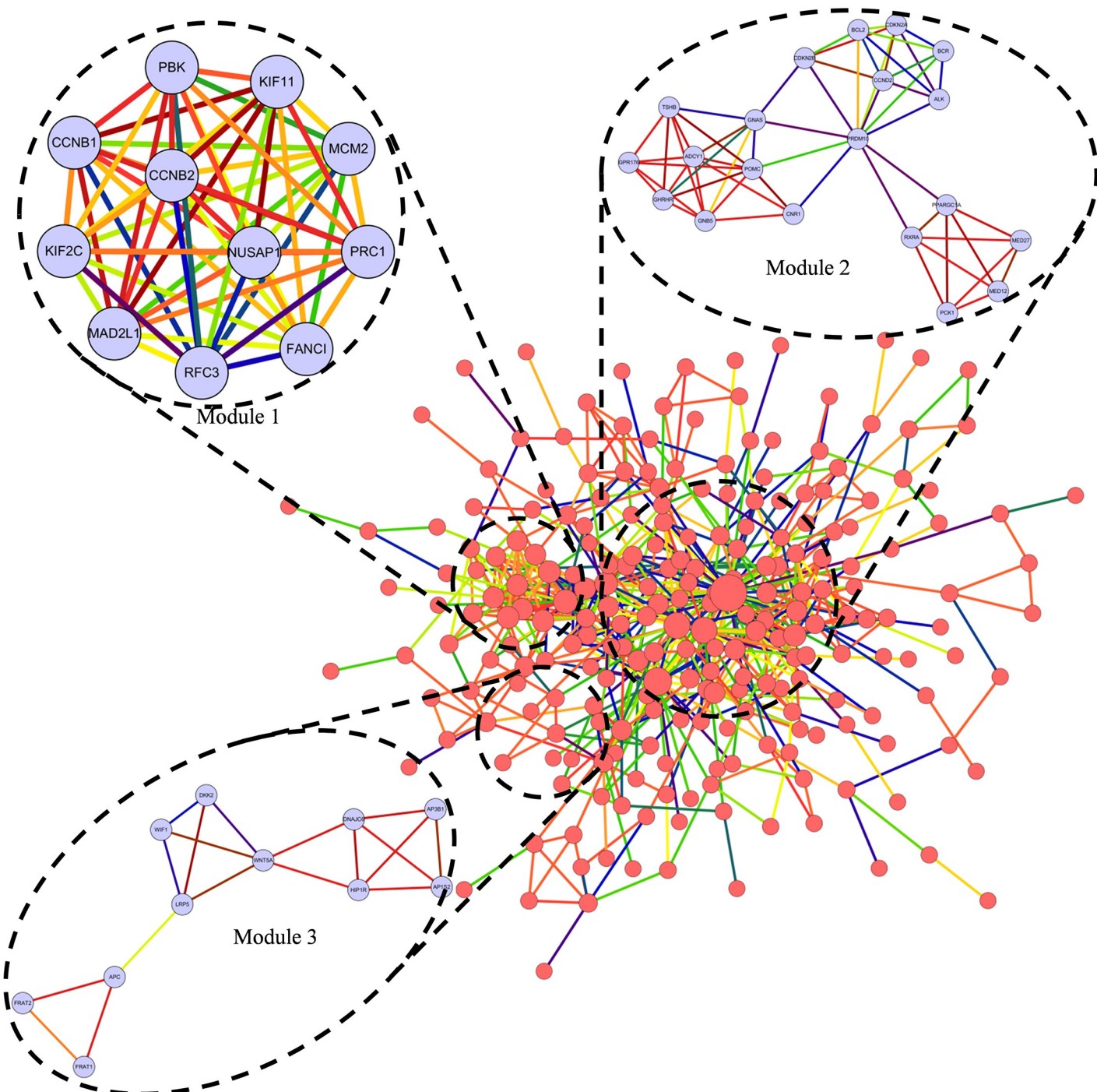

**Fig 3. Top 3 modules from the protein-protein interaction network.**

## STO-609 induces apoptosis of HP75 cells

Flow cytometer was used to analyze the cells treated with different doses of STO-609 to expose the mechanism of STO-609 in NFPA. In the control group, the proportion of normal, necrotic, late apoptosis and early apoptosis was 88.51%, 3.45%, 4.53% and 3.51%, respectively; 66.35%,

**Table 3. The functional annotation and enrichment of modules genes.**

| Module | Term | Count | PValue | FDR | Genes |
|---|---|---|---|---|---|
| module1 | GO:0000922~spindle pole(CC) | 4 | 2.42E-05 | 2.13E-02 | CCNB1, KIF11, MAD2L1, PRC1 |
| | GO:0051301~cell division(BP) | 5 | 3.53E-05 | 3.95E-02 | CCNB1, KIF2C, KIF11, MAD2L1, CCNB2 |
| | GO:0015630~microtubule cytoskeleton(CC) | 4 | 4.80E-05 | 4.22E-02 | KIF2C, CCNB2, PRC1, MCM2 |
| module2 | hsa05200:Pathways in cancer(KEGG) | 8 | 7.41E-05 | 8.34E-02 | ADCY1, CDKN2A, BCR, CDKN2B, RXRA, BCL2, GNB5, GNAS |
| | GO:0071456~cellular response to hypoxia(BP) | 4 | 2.58E-04 | 3.71E-01 | BCL2, PPARGC1A, SLC9A1, PCK1 |
| | GO:0042493~response to drug(BP) | 5 | 5.95E-04 | 8.55E-01 | ADCY1, BCL2, GNAS, PPARGC1A, SLC9A1 |
| module3 | hsa04310:Wnt signaling pathway(KEGG) | 7 | 4.55E-09 | 3.09E-06 | WNT5A, DKK2, FRAT1, FRAT2, WIF1, LRP5, APC |
| | GO:1904886~beta-catenin destruction complex disassembly (BP) | 4 | 2.33E-07 | 2.95E-04 | FRAT1, FRAT2, LRP5, APC |
| | GO:0016055~Wnt signaling pathway(BP) | 5 | 2.97E-06 | 3.76E-03 | WNT5A, DKK2, WIF1, LRP5, APC |

1.66%, 16.46%, and 15.53% in low dose group; 34.81%, 2.58%, 47.84%, and 14.77% in high dose group (Fig 4G and 4H).

## Discussion

Although NFPAs are benign, it is difficult to diagnose and detect them in their early stages. As the disease develops, single surgical resection is difficult to remove the tumor completely. Clinical diagnosis, treatment and prognosis would be significantly improved if the appropriate targets are identified. Few related genes and molecular pathways have been studied by previous researchers, there is an urgent need for comprehensive analysis regarding to NFPAs.

Our study used bioinformatics analysis techniques to screen hub genes and pathways, which provides promising targets for the diagnosis and treatment of NFPA. A previous study pointed that: MYO5A, a deferential expressed gene between invasive and non-invasive NFPA, might be a crucial biomarker for tumor invasiveness [19]. This study used similar bioinformatics methods but combined with more complicated and refined algorithms, to identify possible markers between NFPA and normal pituitary tissues.

We downloaded three datasets: GSE2175, GSE26966 and GSE51618, screened normal tissues and NFPA tissues samples from them, and identified the DEGs between normal tissues and NFPA tissues. The result showed that there were 351 DEGs among which 169 were up regulated and 182 were down regulated. The functions and corresponding signaling pathways of DEGs were investigated by GO, KEGG and GSEA analysis. The results implied that NFPA cells shared similar characteristics of universal cancer cells (cell regulation of growth changed, cell division and ion transport strengthen, the extracellular matrix changed and cell interaction). In addition, DEGs were closely related to dopaminergic synapses, neuronal cell bodies, regulations of cardiac muscle contraction by calcium ion signaling and adrenergic signaling in cardiomyocytes. Besides, GSEA analysis results revealed that NFPAs were closely related to Parkinson's disease (PD) and vibrio cholerae infection. Dopamine agonists were vital medium in medical management of pituitary adenomas and they were also used in the treatment of PD [20]. Previous study also suggested that dopamine inhibited PRL releasing by decreasing intracellular cAMP levels and calcium uptake and reduced the effects of cholera toxin, which

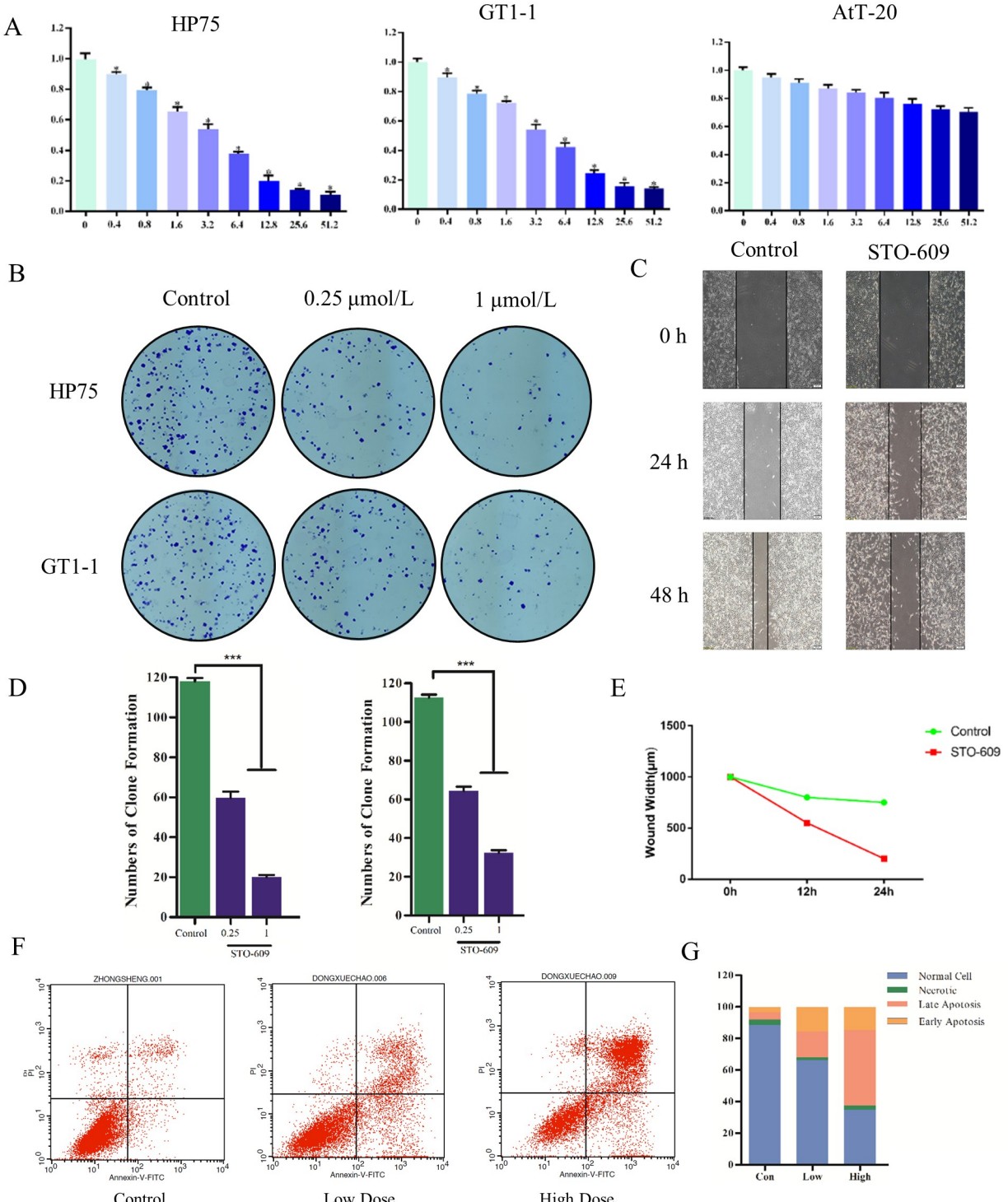

**Fig 4. (A)** Cellular viability of HP75, GT1-1 and AtT-20 cells treated with STO-609 **(B)** Clonogenicities in Petri dishes with different doses of STO-609 **(C)** Scratch assay in control and STO-609 group **(D)** numbers of clone formation in HP75 cell line and GT1-1 cell line **(E)** Wound width in control and STO-609 **(F)** The distribution of cells in apoptosis with different doses of STO-609 **(G)** The percentage of apoptosis cell treated with different dose of STO-609.

stimulated the formation of cAMP [21]. Meanwhile, some pituitary macroadenoma patients complicated with unknown reason movement disorders [22]. Based on the facts above and our results, we hypothesized that NFPA and PD might be homologous. It was decreased dopamine levels in the NFPA patients, which accounted for dopaminergic neuron synapse pathway compensatorily activated in most NFPA patients, that led to PD. In other words, NFPA patients took a rather higher risk suffering PD. However, the detailed mechanisms needed to be elucidated by further studies.

PPI network exposed that four hub proteins encoded by genes (*CALM1*, *PRDM10*, *RIPK4* and *MAD2L1*) had never been reported related to NFPA. *CALM1*, which encoded calmodulin (CaM), played an important role in cell functions such as phosphorylation of tau protein, constitution of various biological membrane structures, signal transduction and synthesis and release of neurotransmitters. It could also affect steroidogenesis mediated by cAMP synthesis [23, 24]. In NFPA cells, *CALM1* was differential expressed significantly. And many abnormal regulated pathways (involved Dopaminergic synapse, cGMP-PKG signaling pathway, Alcoholism, cAMP signaling pathway) were driven by this gene. Evidences also suggested that CALM1 was closely related to neurodegenerative disease such as PD. There were also four important signaling pathways that had been reported about NFPA: MAPK, p53, TGFβ and Jak-STAT [25–28]. These four pathways were closely related to the stimulation of PI3K (phosphoinositide-3-kinase) and the outcome of its product PIP3 (phosphatidylinositol 3,4,5-trisphosphate). Initially, it was CaM that triggered activation of PI3K [29–32]. Accordingly, CALM1 could be considered as the most important therapeutic target for NFPAs. Meanwhile, the $Ca^{2+}$/CaM signaling pathway was aberrantly regulated in NFPAs, which was involved in lots of physiological functions of cells: transcriptional activation, protein synthesis, glycogen metabolism, cell division and so forth. It prompted us to hypothesize that STO-609, a selective inhibitor of CaM-KK, which could block the $Ca^{2+}$/CaM signaling pathway and the function of CALM1, might have anti-NFPAs therapeutic effects and it had also been verified by the following assays in these studies.

*PRDM10*, PR/SET domain 10, was a part of the PRDM (PRDI-BF1 and RIZ homology domain containing) family participating in transcriptional regulation through chromatin remodeling. The protein encoded by *PRDM10* was a transcription factor participating in regulation of transcription and protein binding, which had been reported play a significant role during development of the central nervous system, and in the pathogenesis of neuronal storage disease [33–35]. It is suggested that PRDM 10 might be involved in normal tissue differentiation during mouse embryonic development [35]. There were few reports in the past about PRDM10, and it had not been specially described in any other neoplasm but for undifferentiated pleomorphic sarcoma [33]. In our study, it showed that this gene, which was the core of the interaction with multiple genes, had a pivotal position in NFPA tissues. The expression of this gene was abnormally regulated, in view of *PRDM10* was a driver gene of NFPA, we hypothesized its abnormal activation would promote the initiation of NFPA, it might play roles during the course of NFPA development. PRDM10 might be a biomarker in the early diagnosis of NFPA.

*RIPK4*, receptor interacting serine/threonine kinase 4, encoded a protein which was a serine/threonine protein kinase that interacted with protein kinase C-delta and required for keratinocyte differentiation. RIPK4, the new member in RIP kinase family, might manage NF-kappa B-dependent pro-survival or pro-inflammatory signals negatively. It was reported that this gene was neurodegenerative disease related in a mouse neural stem cells experiment [36], and could also be a novel putative tumor suppressor in human hepatocarcinogenesis [37]. Our study showed that its molecular functions which mainly involved in protein binding, ATP binding, protein serine/threonine kinase activity and protein kinase activity, played important

roles in the proceeding of NFPA. Overexpression of RIPK4 led to activation of JNK and NF-kappa B, which resulted in abnormal cell cycle and proliferation [38]. It was reported that up-regulation of *RIPK4* might help the development of certain tumors, such as skin, ovarian, cervical squamous cell carcinoma, and cervical cancer [39–41]. Our results suggested that *RIPK4* was a driver gene of NFPA, and it might be a novel monitor marker, target or factor for diagnosis, therapy and even prognostic analysis.

*MAD2L1*, mitotic arrest deficient 2 like 1, was part of the mitotic spindle assembly checkpoint that prevented the onset of anaphase until all chromosomes were properly aligned at the metaphase plate. *MAD2L1* jointly contributed to the regulation of mitotic checkpoint and maintained chromosomal integrity [42, 43], interacting with a variety of other molecules such as PTTG1, WT1, AURORA, AURORB and H3K4 [42, 44–46]. Meanwhile, co-depletion of *MAD2L1* and *BUBR1* induced cell cycle arrest and death in addition to aneuploidy [47]. Aneuploidy occurred during tumorigenesis initiation and contributed to tumor formation. Spatial organization of the mitotic regulatory event, and precise timing are vital key in activation of mitotic checkpoint, which ensured accurate chromosome segregation and catalytic activation of the spindle assembly checkpoint depended on regulated protein-protein interactions. *MAD2L1* involved in these processes, the abnormal expression of *MAD2L1* might result in checkpoint control loss, chromosome instability, or early onset of malignancy [43, 48, 49]. In our study, this gene was significantly up-regulated, which revealed that *MAD2L1* was a driver gene of NFPA, as well as a potential a bio-markers for diagnosis and therapy of NFPA.

The effects of STO-609 were evaluated in this study with a series of assays in vitro. The results pointed that the cellular viability in HP75 and GT1-1 cell lines was revealed dose-depended decreased when treated with STO-609. In comparison to control group, clonogenicities of STO-609 group were less and smaller significantly, which showed a consistency with CCK-8 assay. That also pointed out that STO-609 stunted the proliferation of NFPA cells. The scratch assay suggested that the wounds in control group decreased more sharply than that in STO-609 group significantly after 48h. That implied that migration of hypophysoma cells was strongly restrained by STO-609. Moreover, apoptosis assays were calculated by flow cytometer to study the deeper mechanism of STO-609 anti-hypophysoma effect. The result that percentages of apoptosis cells increased as the augment of STO-609 dose revealed that STO-609 could induce the apoptosis of NFPA cells.

## Conclusion

Our study selected out the DEGs and key pathways in the NFPA tissue. The DEGs in this study contribute to synthetic insight of NFPA pathogenesis in molecular level. Four hub proteins encoded by genes: *CALM1*, *PRDM10*, *RIPK4* and *MAD2L1* were aberrantly expressed and could be potentially applied as diagnostic bio-markers, therapeutic targets and prognostic bio-markers. The enrichment functions and pathways were related to $Ca^{2+}$/CaM signaling pathway and parkinson's disease. STO-609, a potent inhibitor regarding to the $Ca^{2+}$/CaM signaling pathway and CALM1, suppressed proliferation and migration of NFPA cells via inducing NFPA cells apoptosis. STO-609 was a promising drug in NFPA treatment.

## Supporting information

**S1 Table. Venn plot analysis results of DEGs among three datasets.**
(XLSX)

## Author Contributions

**Project administration:** Yong Chen.

**Resources:** Yong Chen.

**Supervision:** Yi Wu, Jiajun Dong.

**Validation:** Jiajun Dong.

**Visualization:** Yi Wu.

**Writing – original draft:** Bo Wu, Shanshan Jiang, Xinhui Wang, Sheng Zhong, Yiming Bi, Dazhuang Yi, Ge Liu, Fangfei Hu, Gaojing Dou.

**Writing – review & editing:** Yong Chen, Yi Wu, Jiajun Dong.

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
