## [Decision Letter · Decision Letter 0]

20 Feb 2020

PONE-D-19-34884

Identification of Driver Genes and Key Pathways of Non-functional Pituitary Adenomas Predicts the Therapeutic Effect of STO-609

PLOS ONE

Dear Dr. Chen,

Thank you for submitting your manuscript to PLOS ONE. After careful consideration, we feel that it has merit but does not fully meet PLOS ONE’s publication criteria as it currently stands. Therefore, we invite you to submit a revised version of the manuscript that addresses the points raised during the review process.

We would appreciate receiving your revised manuscript by Apr 05 2020 11:59PM. To enhance the reproducibility of your results, we recommend that if applicable you deposit your laboratory protocols in protocols.io, where a protocol can be assigned its own identifier (DOI) such that it can be cited independently in the future. For instructions see: http://journals.plos.org/plosone/s/submission-guidelines#loc-laboratory-protocols

We look forward to receiving your revised manuscript.

Kind regards,

Tomasz Boczek, Ph.D.

Academic Editor

PLOS ONE

Journal Requirements:

2. Please include your tables as part of your main manuscript and remove the individual files. Please note that supplementary tables (should remain/ be uploaded) as separate "supporting information" files

Reviewers' comments:

Reviewer's Responses to Questions

**Comments to the Author**

1. Is the manuscript technically sound, and do the data support the conclusions?

Reviewer #1: Partly

Reviewer #2: Partly

2. Has the statistical analysis been performed appropriately and rigorously? 

Reviewer #1: Yes

Reviewer #2: No

3. Have the authors made all data underlying the findings in their manuscript fully available?

Reviewer #1: Yes

Reviewer #2: Yes

4. Is the manuscript presented in an intelligible fashion and written in standard English?

Reviewer #1: No

Reviewer #2: No

5. Review Comments to the Author

Reviewer #1: In this manuscript, the authors aim to identify novel molecular pathways and genes regulating non-functional pituitary adenomas (NFPA) which are usually benign tumors, to identify therapeutic targets. They utilize three GEO data sets from gene expression profiling of non-invasive and invasive NFPAs to perform data mining analyses to identify differentially expressed genes, hierarchical clustering, gene ontology, hub gene expression, biological function and pathway analyses. From these analyses, they identify CALM1, PRDM10, and MAD2L1 were recognized as hub genes upregulated in invasive NFPAs while RIPK4 was identified as a downregulated hub gene. Since CALM1, the gene encoding for the intracellular Ca2+ receptor calmodulin (CaM), is one of the upregulated hub genes, the authors utilize STO-609, a selective inhibitor of CaMKKs, the upstream regulator of the CaMK signaling cascade that is regulated by Ca2+/CaM binding, to modulate CaM function in one human NFPA cell line and one mouse pituitary adenoma cell line. They report that treatment of these cells with STO-609 results in a loss of cell viability and migration potential. This is an interesting story, where the data mining and the identification of the four hub genes and a potential similarity to Parkinson disease are interesting. However the manuscript suffers from some very serious flaws:

1. The whole paper is poorly written with bad sentence construction, grammatical and spelling mistakes, wrong word usage and use of colloquial phrases such as “It’s”, “what’s more”, etc. The impact of the potentially exciting discovery of the data mining results they describe for figures 1-3 is lost due to the poor English writing. The authors should extensively edit the paper for English language.

2. There are many abbreviations in this paper, which makes it hard to follow the content at times. Authors should include a table of all the abbreviations used in the paper.

3. The rationale for the STO-609 studies is not clear at all. This should be made in the last paragraph of the introduction as well as in the results section which describes these studies. Moreover, STO-609 does not affect the function of Calmodulin. A statement that STO-609 blocks CALM1 function appears repeatedly in the text in the introduction and discussion without any references to back this claim. This reviewer is not aware of any studies showing this in the literature. If the authors have evidence, they should cite the appropriate papers to back up this claim.

4. Nevertheless, the authors are justified in testing STO-609 against NFPAs. However, this needs to be well-justified.

5. None of the figures are legible. All figures appear blurry, pixelated, and I could not understand what any of them show. Given this, it was hard to judge this paper.

6. Specifically, in figure 4A and B, none of the X-axis values which indicate STO-609 concentrations used for the dose-response curve on cell viability, are legible. So I am not able to judge if the treatment worked or not.

7. The authors do not indicate how STO-609 was solubilized, and prepared for the in vitro assays. They talk about a “solvent DMSO control” in the Materials and Methods section but never show this in Figure 4. Thus, it is hard to understand whether the effects they see of STO-609 on these cells is due to solvent toxicity or STO-609.

8. When performing experiments to understand the effects of a drug on cancer cells, it is important to test if that drug does the same to normal cells. The authors should test STO-609 on normal pituitary cells. There is such a cell line available from the ATCC – AtT-20 mouse normal pituitary cells.

9. Discussion makes several statements and claims without any references. Also, the potential link between Parkinson’s disease and NFPA would be interesting. But since none of the figures are legible, it is not possible to reach this conclusion from the data provided. Moreover, the discussion of this connection should be written in a more focused manner.

10. Table 2: What do the authors mean by the column titled “Betweenness”? This is an incorrect term anyways.

Reviewer #2: Despite that, the subject is very interesting but the presented studies are weakly performed and described. Therefore it is difficult to agree with the authors' conclusions in the present form of the manuscript.

1. Why authors choose only one Ca2+/CaM kinase inhibitor? Its cause that results are very unbelievable.

2. The authors omitted a description of cell treatment, probably, therefore, the concentrations of agent added to the cell culture is not described in the materials and methods.

3. The authors did not describe the surface of the dishes used in the colony-forming assay. Additionally, the results of that experiment are presented unreadable. Therefore it is impossible to assess how many colonies grown up? The authors should present the absolute colony numbers per plate (with plate surface) or per cm2. Did the authors check the percentage of grown colonies in control? If it is less than 50% then the cells are not suitable for this assay. Additionally, the figures (4B) presented cell treated with 0.25 mikrom/L looks similar to 1 mikrom/L. Where did the authors saw the differences presented in the graphs?

4. Similar 4C - the pictures do not presented the results showed in the graphs.

In those reasons demonstrated studies are not convincing and required future analysis that could confirm the authors' supposition.

6. PLOS authors have the option to publish the peer review history of their article (what does this mean?). If published, this will include your full peer review and any attached files.

Reviewer #1: No

Reviewer #2: No

---

## [Author Response · Author response to Decision Letter 0]

5 Aug 2020

Reviewers' comments:

Reviewer 1: 

In this manuscript, the authors aim to identify novel molecular pathways and genes regulating non-functional pituitary adenomas (NFPA) which are usually benign tumors, to identify therapeutic targets. They utilize three GEO data sets from gene expression profiling of non-invasive and invasive NFPAs to perform data mining analyses to identify differentially expressed genes, hierarchical clustering, gene ontology, hub gene expression, biological function and pathway analyses. From these analyses, they identify CALM1, PRDM10, and MAD2L1 were recognized as hub genes upregulated in invasive NFPAs while RIPK4 was identified as a downregulated hub gene. Since CALM1, the gene encoding for the intracellular Ca2+ receptor calmodulin (CaM), is one of the upregulated hub genes, the authors utilize STO-609, a selective inhibitor of CaMKKs, the upstream regulator of the CaMK signaling cascade that is regulated by Ca2+/CaM binding, to modulate CaM function in one human NFPA cell line and one mouse pituitary adenoma cell line. They report that treatment of these cells with STO-609 results in a loss of cell viability and migration potential. This is an interesting story, where the data mining and the identification of the four hub genes and a potential similarity to Parkinson disease are interesting. However the manuscript suffers from some very serious flaws:

The whole paper is poorly written with bad sentence construction, grammatical and spelling mistakes, wrong word usage and use of colloquial phrases such as “It’s”, “what’s more”, etc. The impact of the potentially exciting discovery of the data mining results they describe for figures 1-3 is lost due to the poor English writing. The authors should extensively edit the paper for English language [a].

There are many abbreviations in this paper, which makes it hard to follow the content at times. Authors should include a table of all the abbreviations used in the paper [b].

 The rationale for the STO-609 studies is not clear at all. This should be made in the last paragraph of the introduction as well as in the results section which describes these studies. Moreover, STO-609 does not affect the function of Calmodulin. A statement that STO-609 blocks CALM1 function appears repeatedly in the text in the introduction and discussion without any references to back this claim. This reviewer is not aware of any studies showing this in the literature. If the authors have evidence, they should cite the appropriate papers to back up this claim [c].

Nevertheless, the authors are justified in testing STO-609 against NFPAs. However, this needs to be well-justified [d]. 

None of the figures are legible. All figures appear blurry, pixelated, and I could not understand what any of them show. Given this, it was hard to judge this paper [e].

 Specifically, in figure 4A and B, none of the X-axis values which indicate STO-609 concentrations used for the dose-response curve on cell viability, are legible. So I am not able to judge if the treatment worked or not [f].

 The authors do not indicate how STO-609 was solubilized, and prepared for the in vitro assays. They talk about a “solvent DMSO control” in the Materials and Methods section but never show this in Figure 4. Thus, it is hard to understand whether the effects they see of STO-609 on these cells is due to solvent toxicity or STO-609 [g].

 When performing experiments to understand the effects of a drug on cancer cells, it is important to test if that drug does the same to normal cells. The authors should test STO-609 on normal pituitary cells. There is such a cell line available from the ATCC – AtT-20 mouse normal pituitary cells [h].

Discussion makes several statements and claims without any references. Also, the potential link between Parkinson’s disease and NFPA would be interesting. But since none of the figures are legible, it is not possible to reach this conclusion from the data provided. Moreover, the discussion of this connection should be written in a more focused manner [i].

Table 2: What do the authors mean by the column titled “Betweenness”? This is an incorrect term anyways [j].

Response: Thanks for your comments. We are deeply impressed with your encyclopedic scholarship and conscientious attitude, and we are honored to get your comments.

a, Thanks for your professional comments. This manuscript was polished by a native English speaker, and inopportune expressions was corrected.

b, We had provided the Abbreviations document. Thanks for your comments.

c, Thanks for your professional comments. As we described in this manuscript, STO-609 was an inhibitor of Ca2+/calmodulin-dependent protein kinase kinase (CaM-KK2), which is an important downstream role of Calmodulin. So, STO-609 affect the function of Calmodulin indirectly. The statement that STO-609 blocks CALM1 function in this manuscript was expressed more accurate, and the rationale for the STO-609 studies also was made more clear in the last paragraph of the introduction as well as in the results section which describes these studies. Thanks for your comments again.

d, In order to verify the anti-NFPAs effects of STO-609, we performed a series of assays such as CCK-8 assay, Colony-forming assay, in vitro scratch assay and apoptosis assays. CCK-8 and colony-forming assay were conducted to estimate whether STO-609 inhibited the proliferation of NFPA cells, while in vitro scratch assay could demonstrate whether STO-609 reduced the migration of NFPA cells. In addition, apoptosis assays revealed how STO-609 induced the apoptosis of NFPA cells. These experimental methods were reliable and widely used in studying the effects of drugs on cells. For example, Zhang W, Lei Z etc performed CCK-8, colony-forming assay and cell scratch assay to research the effect of water extract of sporoderm-broken spores of ganoderma lucidum on osteosarcoma cells (Zhang W, Lei Z, Meng J, Li G, Zhang Y, He J, Yan W. Water Extract of Sporoderm-Broken Spores of Ganoderma lucidum Induces Osteosarcoma Apoptosis and Restricts Autophagic Flux. OncoTargets and Therapy 2019;12:11651-11665. doi: 10.2147/OTT.S226850.). The details of assays in vitro were provided and Figures was polished according to your following comments. Thanks for your professional comments.

e, Sorry for blurry figures. All the figures was replaced with clear PDF versions.

f, Sorry for that. We optimized the format of Figure 4, and made it easier to read. Thanks for your comments. 

g, Thanks for your comments. Firstly, we dissolved STO-609 in pure DMSO to obtain the mother liquor, then we respectively added appropriate culture medium to the mother liquor to configure cell culture medium with STO-609 concentration of 0.25μmol/L and 1μmol/L. The concentration of DMSO was less than 0.5%. In this concentration, the influence of DMSO on cells was negligible, therefore we decided not to set up the DMSO control group. This statement was added to the manuscript.

h, Thanks for your comments. The ATCC – AtT-20 mouse normal pituitary cells were tested in this manuscript, and the results were added in this manuscript.

i, Thanks for your professional comments. The reference you mentioned was added in discussion. In this study, GSEA analysis was performed for further insight of DEGs, and the results showed that the pathway of Parkinson’s disease (PD) was significantly altered. That implicit the potential link between PD and NFPA. Thus, we hypothesized that NFPA and PD might be homologous and NFPA patients may take a rather higher risk suffering PD. This hypothesis needed further studies to elucidated in future. 

j, Thank you for pointing out our mistake. We changed “Betweenness” to “Betweenness Centrality” in Table 2. 

Reviewer 2: 

 Despite that, the subject is very interesting but the presented studies are weakly performed and described. Therefore it is difficult to agree with the authors' conclusions in the present form of the manuscript.

 Why authors choose only one Ca2+/CaM kinase inhibitor? Its cause that results are very unbelievable [a]. 

 The authors omitted a description of cell treatment, probably, therefore, the concentrations of agent added to the cell culture is not described in the materials and methods [b].

 The authors did not describe the surface of the dishes used in the colony-forming assay. [c] Additionally, the results of that experiment are presented unreadable. Therefore it is impossible to assess how many colonies grown up? The authors should present the absolute colony numbers per plate (with plate surface) or per cm2. Did the authors check the percentage of grown colonies in control? If it is less than 50% then the cells are not suitable for this assay. [d] Additionally, the figures (4B) presented cell treated with 0.25 mikrom/L looks similar to 1 mikrom/L. Where did the authors saw the differences presented in the graphs? [e] Similar 4C - the pictures do not presented the results showed in the graphs [f].

In those reasons demonstrated studies are not convincing and required future analysis that could confirm the authors' supposition.

Response: It’s our honor to obtain your comments, and thank you for your excellent comments that benefit us a lot. 

a, Thanks for your comments. This study aims to provide potential biomarkers and drug targets for diagnosis and treatment of NFPAs and verify the anti-NFPAs effects of STO-609 preliminary. Actually, we compared all Ca2+/CaM kinase inhibitors, but chose STO-609 finally because STO-609 had been proved playing roles in several diseases recently such as prostate cancer, gastric carcinoma, et al. Thanks for your comments again.

b, Thanks for your comments. The description of STO-609 concentrations was added in this manuscript.

c, Thanks for your professional comments. In colony-forming assay, we used a six-well cell culture plate rather than petri dishes, and the surface area of each well of the culture plate was 9.6 cm2. This description was added in this manuscript. 

d, Thanks for your professional comments. We reanalyzed the results of colony-forming assay, and added the absolute colony numbers (with plate surface) to the manuscript and Figure 4. In addition, the percentage of grown colonies in control group was 70%.

e, The figure and graphs were replaced with clear PDF versions. Thanks for your comments which benefit us a lot.

f, Thanks for your comments. The figure was polished.

---

## [Decision Letter · Decision Letter 1]

15 Sep 2020

PONE-D-19-34884R1

Identification of Driver Genes and Key Pathways of Non-functional Pituitary Adenomas Predicts the Therapeutic Effect of STO-609

PLOS ONE

Dear Dr. Chen,

Thank you for submitting your manuscript to PLOS ONE. After careful consideration, we feel that it has merit but does not fully meet PLOS ONE’s publication criteria as it currently stands. Therefore, we invite you to submit a revised version of the manuscript that addresses the points raised during the review process.

We look forward to receiving your revised manuscript.

Kind regards,

Tomasz Boczek, Ph.D.

Academic Editor

PLOS ONE

Reviewers' comments:

Reviewer's Responses to Questions

**Comments to the Author**

1. If the authors have adequately addressed your comments raised in a previous round of review and you feel that this manuscript is now acceptable for publication, you may indicate that here to bypass the “Comments to the Author” section, enter your conflict of interest statement in the “Confidential to Editor” section, and submit your "Accept" recommendation.

Reviewer #1: (No Response)

Reviewer #2: (No Response)

2. Is the manuscript technically sound, and do the data support the conclusions?

Reviewer #1: Yes

Reviewer #2: (No Response)

3. Has the statistical analysis been performed appropriately and rigorously? 

Reviewer #1: (No Response)

Reviewer #2: No

4. Have the authors made all data underlying the findings in their manuscript fully available?

Reviewer #1: Yes

Reviewer #2: Yes

5. Is the manuscript presented in an intelligible fashion and written in standard English?

Reviewer #1: Yes

Reviewer #2: No

6. Review Comments to the Author

Reviewer #1: The manuscript has improved a lot with the revision. A few minor concerns remain which need to be addressed.

1. Pages 5-6, lines 128-129: - This sentence needs to be revised. The reference # 49 does not show that STO-609 blocks CALM1 function. You could say, for instance, that you wanted to use STO-609 to block CaMKK signaling, as it operates downstream of CALMA. Please understand that calmodulin activates many proteins, and CaMKK is only one of them.

2. Page 7 line 177 - Delete the part of the sentence that says STO-609 blocks CALM1 gene function. It does not. Also you need to say here how you made the STO-609 solution.

3. Page 8, line 197-198: - Change "mother liquor" to stock solution. The description of STO-609 stock preparation should be moved earlier to page 7 line 177 - first time you mention about using STO-609 on cells.

Page 8, lines 199-202: I disagree. Even 0.05% DMSO could have an effect on cells. This is an important control. Also, Remove the sentence: "different doses of STO-609 were used to treat those cells".

Page 11, lines 268 -276: It is still not clear why they chose to block CaMKK function using STO-609 while PRDM10 scored the highest.. Please make this clear. You do have a paragraph in the Discussion - page 14 lines 348-350 that provide a good rationale. This can be utilized in the Introduction and the Results sections to provide the rationale for blocking CaMKK.

Pages 11-12, lines 281-282: - Revise these sentences: Meanwhile, only tiny decline was observed in STO-609 group. What’s more, the wounds in the drug group were quite wider than in control group after 48h.

Reviewer #2: Unfortunately, despite the author's corrections, the manuscript still has low quality. The authors' explanation about the reason for choosing the only one inhibitor in the present studies is very unclear. The conclusions from such limited analysis are dubious. The quality of figures and language make reading the text difficult. The result of wound healing is still inconsistent (graphs and images) - Figure 4.

To sum up, I regret to inform you that I do not recommend the manuscript in the present form for publication in PLOS ONE.

7. PLOS authors have the option to publish the peer review history of their article (what does this mean?). If published, this will include your full peer review and any attached files.

Reviewer #1: No

Reviewer #2: No

---

## [Author Response · Author response to Decision Letter 1]

21 Sep 2020

1. If the authors have adequately addressed your comments raised in a previous round of review and you feel that this manuscript is now acceptable for publication, you may indicate that here to bypass the “Comments to the Author” section, enter your conflict of interest statement in the “Confidential to Editor” section, and submit your "Accept" recommendation.

Reviewer #1: (No Response)

Reviewer #2: (No Response)

2. Is the manuscript technically sound, and do the data support the conclusions?

Reviewer #1: Yes

Reviewer #2: (No Response)

3. Has the statistical analysis been performed appropriately and rigorously?

Reviewer #1: (No Response)

Reviewer #2: No

Response: The statistical analysis was updated. Independent-samples t test was conducted to analyze quantitative data of bioinformation analysis, while Analysis of Variance (ANOVA) was performed to analyse multiple comparison data of cell experiments. Besides, we conducted Dunnett-t test as post hoc test after ANOVA. We confirmed that the level of significance was P < 0.05. Thanks a lot for the valuable comments.

4. Have the authors made all data underlying the findings in their manuscript fully available?

Reviewer #1: Yes

Reviewer #2: Yes

5. Is the manuscript presented in an intelligible fashion and written in standard English?

PLOS ONE does not copy edit accepted manuscripts, so the language in submitted articles must be clear, correct, and unambiguous. Any typographical or grammatical errors should be corrected at revision, so please note any specific errors here.

Reviewer #1: Yes

Reviewer #2: No

Response: Thanks for the professional comments. We had already re-polished the language in this manuscript to meet the requirements of PLOS ONE.

6. Review Comments to the Author

Reviewer #1: The manuscript has improved a lot with the revision. A few minor concerns remain which need to be addressed.

1.Pages 5-6, lines 128-129: - This sentence needs to be revised. The reference # 49 does not show that STO-609 blocks CALM1 function. You could say, for instance, that you wanted to use STO-609 to block CaMKK signaling, as it operates downstream of CALMA. Please understand that calmodulin activates many proteins, and CaMKK is only one of them.

Response: Thanks a lot for your detailed comments. This sentence was revised according to this suggestion.

2.Page 7 line 177 - Delete the part of the sentence that says STO-609 blocks CALM1 gene function. It does not. Also you need to say here how you made the STO-609 solution.

Response: This part was revised according to your comments. Thank you for the valuable suggestions. 

3. Page 8, line 197-198: - Change "mother liquor" to stock solution. The description of STO-609 stock preparation should be moved earlier to page 7 line 177 - first time you mention about using STO-609 on cells.

Page 8, lines 199-202: I disagree. Even 0.05% DMSO could have an effect on cells. This is an important control. Also, Remove the sentence: "different doses of STO-609 were used to treat those cells".

Page 11, lines 268 -276: It is still not clear why they chose to block CaMKK function using STO-609 while PRDM10 scored the highest.. Please make this clear. You do have a paragraph in the Discussion - page 14 lines 348-350 that provide a good rationale. This can be utilized in the Introduction and the Results sections to provide the rationale for blocking CaMKK.

Pages 11-12, lines 281-282: - Revise these sentences: Meanwhile, only tiny decline was observed in STO-609 group. What’s more, the wounds in the drug group were quite wider than in control group after 48h.

Response: These sections were all revised with your comments. Besides, the point of view that the influence of 0.1% DMSO on cells is negligible is supported by researches as following:

[1]Qi W, Ding D, Salvi R J. Cytotoxic effects of dimethyl sulphoxide (DMSO) on cochlear organotypic cultures[J]. Hearing Research, 2008, 236(1-2):52-60

[2]Nirogi R, Kandikere V, Bhyrapuneni G, et al. Effect of dimethyl sulfoxide on in vitro cytochrome P4501A2 mediated phenacetin O-deethylation in human liver microsomes[J]. Drug Metabolism & Disposition the Biological Fate of Chemicals, 2011, 39(11):2162

 These suggestions benefited us a lot in improving this manuscript. It is our great honor to obtain your valuable and professional comments.

Reviewer #2: Unfortunately, despite the author's corrections, the manuscript still has low quality.[a] The authors' explanation about the reason for choosing the only one inhibitor in the present studies is very unclear. The conclusions from such limited analysis are dubious.[b] The quality of figures and language make reading the text difficult.[c] The result of wound healing is still inconsistent (graphs and images) - Figure 4.[d] 

To sum up, I regret to inform you that I do not recommend the manuscript in the present form for publication in PLOS ONE.[a] 

Response: a, In this study, bioinformatical methods were conducted to identify DEGs, comprehensively investigate hub genes, annotate enrichment functions and key pathways of NFPAs, and performed a series of cell assays to verify the therapeutic effect of STO-609 on NFPAs. The bioinformatical methods and cell assays in this study were scientific and reliable, which were widely used in researches as following:

[1]Zhong S, Wu B, Li J et al. T5224, RSPO2 and AZD5363 are novel drugs against functional pituitary adenoma [J]. Aging, 2019;11(20):9043-9059.

[2]Zhong S, Wu B, Han Y et al. Identification of Driver Genes and Key Pathways of Pediatric Brain Tumors and Comparison of Molecular Pathogenesis Based on Pathologic Types [J]. World Neurosurgery, 2017;107:990-1000.

[3]Shen S, Kong J, Qiu Y et al. Identification of core genes and outcomes in hepatocellular carcinoma by bioinformatics analysis [J]. Journal of Cellular Biochemistry, 2019;120(6):10069-10081.

b, Thanks for your comments. This study aims to provide potential biomarkers and drug targets for diagnosis and treatment of NFPAs and verify the anti-NFPAs effects of STO-609 preliminary. After comparing all Ca2+/CaM kinase inhibitors, we found STO-609 was a selective and cell-permeable inhibitor of the CaM-KK, and was proved playing roles in prostate cancer, gastric carcinoma, et al. We conjectured STO-609 was also a potential drug for NFPAs, therefore we regarded STO-609 as the major object in this research. Other Ca2+/CaM kinase inhibitors were outside the scope of this study. We believe other Ca2+/CaM kinase inhibitors could be interesting directions in further researches. 

c, This language in this manuscript was polished to meet the requirements of PLOS ONE. Besides, we improved the resolution of figures to make them more clear for publication. Thanks for your comments.

d, We increased the images resolution of the microscopy images in Fig.4 and believed that this version was clear enough to compare the widths of scratch. Thank you for the useful suggestions.

Dear reviewers, 

Thanks sincerely for your hard work dealing with this manuscript. These professional comments and suggestions really helped us a lot to improve the quality of this manuscript. We believe these opinions will benefit us on our researches in the future. 

If there is any problem, please don’t hesitate to contact us. We will reply you as soon as possible. Best wishes to you and all whom you love.

Best regards,

Yong Chen

---

## [Editor Report · Decision Letter 2]

23 Sep 2020

Identification of Driver Genes and Key Pathways of Non-functional Pituitary Adenomas Predicts the Therapeutic Effect of STO-609

PONE-D-19-34884R2

Dear Dr. Chen,

We’re pleased to inform you that your manuscript has been judged scientifically suitable for publication and will be formally accepted for publication once it meets all outstanding technical requirements.

Kind regards,

Tomasz Boczek, Ph.D.

Academic Editor

PLOS ONE
---

## [Editor Report · Acceptance letter]

8 Oct 2020

PONE-D-19-34884R2 

Identification of Driver Genes and Key Pathways of Non-functional Pituitary Adenomas Predicts the Therapeutic Effect of STO-609 

Dear Dr. Chen:

I'm pleased to inform you that your manuscript has been deemed suitable for publication in PLOS ONE. Congratulations! Your manuscript is now with our production department. 

Kind regards, 

on behalf of

Dr. Tomasz Boczek 

Academic Editor

PLOS ONE